# Liking and Description of Pasta Sauces with Varying Mealworm Content

**DOI:** 10.3390/foods12173202

**Published:** 2023-08-25

**Authors:** Marlies Wallner, Nina Julius, Raquel Pelayo, Christina Höfler, Simon Berner, René Rehorska, Lisa Fahrner, Susanne Maunz

**Affiliations:** 1Institute of Dietetics and Nutrition, University of Applied Sciences FH JOANNEUM, 8020 Graz, Austriachristina.hoefler@fh-joanneum.at (C.H.); susanne.maunz@fh-joanneum.at (S.M.); 2Institute of Applied Production Sciences, University of Applied Sciences FH JOANNEUM, 8020 Graz, Austria; simon.berner@fh-joanneum.at (S.B.); rene.rehorska@fh-joanneum.at (R.R.);; 3Institute of Food and Beverage Innovation, Zurich University of Applied Sciences, 8820 Wädenswil, Switzerland; julu@zhaw.ch

**Keywords:** mealworm, acceptance testing, CATA, ideal product, neophobia, oral disgust

## Abstract

Entomophagy is directly connected with culture, explaining why it is commonly rejected in Western countries. Due to increased meat consumption in recent years with its associated negative impacts on health and sustainability, the development of products based on alternative protein sources has become urgent. The larval form of *Tenebrio molitor* (mealworm) has the potential to substitute meat as it requires less resources and produces less emissions compared to other forms of meat production. Therefore, in this project we have aimed to develop pasta sauces with differing mealworm contents based on a common meat sauce and to test the acceptance with 91 consumers in Austria. Three sauces (100% mealworm, 50% mealworm and 50% meat, 100% meat) were developed and tested using a 9-point hedonic scale for acceptance, and the CATA (Check-All-That-Apply) method was integrated to also receive descriptive information. The analysis of the liking data revealed that the liking for the hybrid sauce with meat and mealworm content was comparable to the meat sauce (6.9 ± 1.8. vs. 6.5 ± 1.8, *p* > 0.05). Less liked was the sauce with the highest mealworm content (5.7 ± 1.8, *p* < 0.05). The CATA analysis demonstrated the strongest positive effects on the mean in terms of how much the products were liked for the attribute “fleshy” (0.8). On the other hand, the attributes “brownish” (−0.9) or “mushy” (−1.0) had the strongest negative effects on the mean of the liking of products. We have seen that meat cannot be substituted by mealworm immediately and completely. The results suggest a stepwise substitution and the further adaptation of products regarding the (negative and positive effecting) attributes to increase consumer acceptance.

## 1. Introduction

The consumption of insects (entomophagy) is a tradition in many countries worldwide, mainly in North and South America, Africa, Asia, and some Pacific Islands [1]. It is known that eating insects is directly connected with these cultures, explaining why entomophagy is commonly rejected in Western countries due to unfamiliarity with the practice [1]. 

In Europe, leading countries of the edible insects market (in 2019) were the United Kingdom, the Netherlands, and France [2,3]. In the same year, revealed by the Global Market Insight report, Europe was in second place, with a 21% share of the edible insects market in terms of revenue [2,3] (following the Asian-Pacific region of 41%). Since 2021, the insect species *Tenebrio molitor*, *Locusta migratoria*, and *Acheta domesticus*, respectively, have been authorized for human consumption in the EU. They are regulated by the Novel Foods (NFs) Regulation (EU) 2015/2283 [2], which regulates food that has not been consumed significantly prior to May 1997, according to EU regulations [4].

Nutritional habits have changed in past years worldwide, and the consumption of meat has increased from 23.1 kg in 1961 to 42.2 kg per person per year in 2011, almost double the quantity [5]. More data provided by the Food and Agriculture Organization of the United Nations (FAO) database showed data from 2011 to 2020, where these numbers were still high and increasing yearly. The meat quantity supplied worldwide has increased from 41.41 kg/capita/year in 2011 to 42.76 kg/capita/year in 2020 [6]. The Global Market Insights report also estimated an increase of 47% between 2019 and 2026 in the edible insects market [2]. 

One of the Sustainable Development Goals (SDGs) of the United Nations is to increase food production by 70% in order to feed the world in 2050, while at the same time a significant decrease in environmental impact is required [7,8]. In this context, one option is to reduce meat consumption and implement entomophagy into the Western diet as there are advantages for both health and sustainability. 

Some of the advantages in the production of insects compared to livestock are the reduced need of land and water for production, the reduction in greenhouse gases and carbon dioxide emissions as well, lowering the environmental impact [7]. 

Nevertheless, insects are not completely harmless; their food safety are conditioned by different aspects. Taking into account whether the insects are reared in the wild or controlled, the conditions under the insects are processed, as well as their environment and other factors [9]. It is important to consider pesticide residues, concentrations of heavy metals, as well as microbiological safety [10,11]. 

In addition, there are some benefits regarding the nutritional value of insects in comparison to meat, demonstrating their potential for usage in the human diet and food industry. As reported by Orkusz (2021) [12], adult forms of *Tenebrio molitor* (24.13 g/100 g) and the larval form (25 g/100 g) were 3.78% and 4.65%, respectively, higher in protein content than meat ((19.2–21.5 g/100 g) turkey breast and chicken breast amongst other types of meat evaluated). Further studies revealed that *Tenebrio molitor* presents a favorable amino acid and a diverse fatty acid composition with a high lipid content (27.4 g/100 g), a good polyunsaturated fatty acid (PUFA) and saturated fatty acid (SFA) ratio (higher than 0.40), contains omega 3 and 6 essential polyunsaturated fatty acids, as well as having a sufficient vitamins and minerals profile [13,14,15,16,17]. Nevertheless, it is important to consider the bioavailability of nutrients in edible insects also with regard to their chitin content [18].

Among all the meat analogues that are becoming promoted to reduce meat consumption, some comparisons revealed that plant-based products were the most accepted, followed in second place by single-cell-based protein. The least liked for the population were insects and cultured meat [19]. In addition, still in 2019 in Western countries, consumers considered insects in their diet to be disgusting [20].

The acceptance and neophobia (the fear of eating novel foods) of eating insects were determined by an internet-based questionnaire in the study of Wendin et al. It was shown that eating foods with protein insect powder was more widely accepted among the participating consumers than food with whole insects [21]. Whilst distinguishing the acceptance of eating invisible/visible insects among non-neophobic and neophobic volunteers, it was revealed that the non-neophobic group were more willing to eat invisible insects (4.52 ± 2.13) than visible ((2.41 ± 1.79), *p* < 0.05). In general, they are more positive about entomophagy in comparison with neophobic persons regarding the consumption of invisible versus visible insects, respectively (2.36 ± 1.78 and 1.50 ± 0.76, *p* < 0.05). 

Furthermore, a sex difference was revealed in this study as men are more positive about the consumption (3.07) and the purchasing (3.04) of visible insects compared to women (1.98 and 2.3, *p* < 0.05) [21].

And women can be considered as more neophobic in this context than men because the latter eat and buy visible insects as food more often than women (*p* < 0.005) [21].

Also, Caparros Megido et al. (2016) published a study where Belgian women and men tried different burgers made of beef, lentils, mealworms and beef, and mealworms and lentils. Some of the conclusions they arrived at were that people in Western countries are not ready yet to integrate whole insects in their menus as they are more willing to eat them with less visibility. Because of that, a transitional period is needed for people to integrate insects in their daily diets [22]. Therefore, disguising or masking the insect as a food source is the main focus in the product development of insect-based foods.

Although, the most common form of consumption is still to eat them as a whole insect in roasted, fried, or boiled form. To encourage more consumers to integrate insects in their diet, they need to be incorporated into other food products in smaller particle size, e.g., dried and ground [23,24]. The food industry is creating new foods based on smaller particles or totally invisible insects and trying to copy the characteristics of traditional meat products [25].

A common psychological pathway to increase the liking of a new food is to combine it with a known one. It was shown that masking insects in familiar products increases the willingness to try them because the anticipation is good [25,26]. Tan et al. (2015) found that insect eaters and non-eaters (either from Thailand or the Netherlands) were more willing to try insect foods when these are invisible in the product (such as grounded) and more similar to typical foods than when they are just covered or completely recognizable [25].

Pliner and Stallberg-White (2000) showed in their study how familiar flavors increase children’s willingness to try novel foods. They experimented with the combination of familiar and novel chips mixed with different types of dips: a previous exposed dip, an unexposed one, an already familiar dip, and an incongruous but familiar dip. From this paper, different conclusions were obtained, and to assess the willingness a 5-point scale was used. Regarding just the novel food and their combinations, it was discovered that there was more willingness to try the novel food when it was combined with an already familiar dip with a rating value of (3.31 ± 1.26) than with an unexposed dip (2.94 ± 1.08) or an incongruous dip (3.06 ± 0.88) [27]. Also, it was observed that in every condition, the participants were more willing to try the familiar food than the novel food [27]. 

Following some of the conclusions obtained above, it was considered appropriate to combine the mealworms with an already known food. Based on this, we tried to imitate a typical meat sauce with pasta to include the insects as a more attractive option. The mealworms were integrated into a pasta sauce, a product that is generally more likely to be consumed. The aim of the present study was to develop a pasta sauce with varying mealworm contents and evaluate them for acceptance and sensory properties (appearance, odor, taste, mouthfeel) among consumers.

## 2. Materials and Methods

### 2.1. Participants

Within this study, 170 adults aged 18–60 years were recruited via social media and around university to fill in the survey; out of these, 91 participated in the real tasting. Persons with an allergy or intolerance or pregnant women were excluded because of the risk. Written consent was obtained from all participants before the testing, which was approved prior to the study’s onset by the ethical committee of the Medical University of Graz (No. 34-081 ex 21/22) and prepared according to the Declaration of Helsinki. 

### 2.2. Testing Procedure

The study was planned in three steps: First, a pasta sauce with different mealworm concentrations was developed. Then, a questionnaire was prepared on background variables, including insect specific contents. And, third, the tasting was performed. Here, the focus lies on the acceptance of the pasta sauces as real products with insects, and testing the consumer acceptance was the main objective of the study.

Pasta sauces were prepared max. 2 weeks ahead of the tastings; they were stored in the freezer and defrosted overnight in the fridge. The samples were presented in small glasses (20 mL) and numbered with a specific and randomized 3-digit code. 

On the day of tasting, the samples were heated in a water bath to 60 °C until tasting. The tasting room and the preparation room were disinfected and ventilated according to hygienic protocol established due to the COVID-19 pandemic. 

The tasting was conducted in the sensory booths at the Health Perception Lab of the FH JOANNEUM, University of Applied Sciences (Graz, Austria).

Consumers were instructed on how to use the online questionnaire and the tasting. Then, the samples were presented in a monadic sequential order, and for neutralization, a half slice of white bread and a glass of water at room temperature were served. 

### 2.3. Development of Pasta Sauces

For the study, three different sauces were developed. All of them were based on the idea of a tomato meat sauce, which has received high liking among many meat-eating consumers. This was also validated by the statistics where tomato has been established as the most consumed vegetable in Austria in 2020–2021 [28]. So, one prepared sauce with mealworms was called “Mealworm sauce”, the second sauce was with minced meat (mixed of 50% pork and 50% beef meat) and called “Meat sauce”, and the third sauce was made of mealworms and minced meat (with the same quantity of both protein sources), so it was called “Hybrid sauce”. 

The procedure to make the mealworms edible was to process them to a “minced mealworm dough” and use this in the same that way minced meat is normally used to cook a Bolognese sauce. However, to achieve this result, pre-treatment of the mealworms was necessary to ensure the quality and safety of the sauces.

Firstly, the procedure to kill, sterilize, and blanch the mealworms was to put them in boiling water for 3 min. Subsequently, they were deep-frozen and stored and ready to be used. Secondly, the defrosted worms were cut and mixed with a dough made of regional products such as crushed green spelt, beetle bean meal, and hesperide vinegar. Thus, the consumption of local products, contributing to a more sustainable system, was promoted. The dough obtained was passed through a meat grinder to obtain a similar appearance and texture to minced meat (Figure 1). Finally, the minced mealworm dough and minced meat were processed in a food processor with a stable temperature and stirring abilities. Some other ingredients were added to obtain the final three sauces, such as mushrooms, oil, celery, carrots, onions, canned and concentrated tomato, and spices. Mushrooms were added with a specific purpose; at first, the intention was to improve and cover the mouthfeel that is created by mealworms. But it was also noticed that this incorporation helped the consistency and juiciness of the sauce. Moreover, as mushrooms have a characteristic taste, it was considered appropriate to add them to all the sauces to make them more similar to each other. 

All the ingredients that were used for the recipes were natural; therefore, no food preservatives were involved in the cooking process.

Regarding the nutritional value of the sauces, it has to be clarified that as meat and mealworms have different protein contents (Table 1), the sauces are going to have differing protein content as well. 

In Table 2, the percentages of protein content of the final recipes (Meat sauce, Hybrid sauce, and Mealworm sauce) are collected and compared. The amount of minced meat and mealworms used was not the same because mealworms are more difficult to manipulate during the cooking process. This is why only a 20% content of mealworms (in 100% of the final Mealworm sauce recipe) was used in contrast to the 31% content of minced meat that was used (in 100% of the Meat pasta sauce) [15]. Furthermore, the total content of meal derived from *T. molitor* larvae should not exceed 20% due to an increase in the pH value and associated shelf-life issues. In addition, sensory quality losses may also occur [29,30].

On the other hand, some of the tomato meat sauces that are now in the supermarkets in Austria have around a 19% beef and pork content in total, and in vegan options, it is around 5% soya protein. In comparison with the sauces developed, the percentage of meat and/or mealworms added was quite similar or higher to these commercialized tomato sauces. The mealworm sauce contains 20% mealworms, the hybrid version of the sauce has 16.6% animal content (15.5% minced meat and 3.1% mealworms), and the meat sauce contains 31% meat (beef and pork). Thus, it is observed that the common percentages used were conserved or exceeded.

### 2.4. Questionnaires

A questionnaire was requested to be filled in by the participants at least one week prior to tasting the sauces to exclude any impact on the results. The questionnaire contained questions on demographics, lifestyle variables, food neophobia scale, and the disgust scale.

Within the demographic and lifestyle section, sex (male, female, other, no answer), age, and activity levels were asked.

At the outset, questions relating more to insects, the familiarity, and regularity of insect consumption and general eating attitudes were requested (“Did you ever taste insects before?” yes/no, “Which insects did you consume?” mealworms/other, “Do you like eating insects?” yes/no, “How often do you consume insects/meat and sausages/plant based meat alternatives?” daily/4–6 times a week/1–3 times a week/1–3 times a month/rare/never, “If never, are you vegetarian or vegan? vegetarian/vegan).

Furthermore, with the food neophobia scale [31], 10 items on a 7-point scale were scored ranging from “do not agree at all” to “totally agree”. 

To receive information on individual disgust sensitivity and especially on oral disgust, a validated questionnaire [32] was used. The questionnaire was validated in the German language and did not require translation. Statements like “You try to eat monkey meat.” or “You see someone throwing up” should be evaluated with regard to their own disgust on a 5-point scale ranging from “not disgusting” to “very disgusting”. 

### 2.5. Sensory Analysis 

Untrained consumers tested the pasta sauces in the sensory booth of the HPL (Health Perception Lab). The acceptance testing was conducted using a 9-point hedonic scale for the testing of overall acceptance and flavor/aroma of the samples. The scale ranged from score 1, “dislike extremely”, to 9, “like extremely”.

The CATA method was used to obtain specific information on consumer perception. Terms were collected and tested prior to consumer analysis in a smaller group of trained and project members (*n* = 6). Attributes were collected via literature research, translated to German, and discussed during the tasting sessions. From this list, the most relevant attributes were used for the main consumer testing task. The terms were presented in randomized order.

In the consumer test, the CATA method was used for the three products and for an ideal product. By collecting the attributes for the ideal product, it was possible to create a sensory profile that most likely met the tasters’ expectations.

### 2.6. Statistics

Statistical analyses of all the experimental data (except for the ranking test involved in sensory evaluation) obtained were performed using SPSS (Statistical Package for the Social Sciences Version 17.0, SPSS Inc., Chicago, IL, USA), at one-way analysis of variance (one-way ANOVA), and Tukey’s post hoc Honestly Significant Difference (HSD) test at a significance level *p* < 0.05. All the data were reported using mean values of the triplicate determinations ± standard deviation. The significant differences among the ranking data (sensory evaluation) were evaluated using the non-parametric Friedman test followed by Fisher’s Least Significant Difference (LSD) test, both *p* < 0.05. 

Statistical Analysis for the acceptance test (ANOVA and Tukey’s HSD) and CATA Analysis (Corchan’s Q- Test and Mc Nemar, penalty analysis, and the principal component analysis involved in sensory evaluation) were performed using XLSTAT (16.0.15225 (64 bit) ver. 22.2.1 for Windows 10).

## 3. Results

### 3.1. Participants Charcteristics

Out of the 170 participants who filled in the questionnaire, 91 participants came to the tasting; of these, there were 51 males and 40 females. The groups of tasting attendants were identified as less food neophobic, had a lower disgust overall mean, a lower oral disgust, and a higher score for acceptance of entomophagy. Furthermore, they were asked their preference of intensive foods, as these types of foods are more often associated with lower liking. We found a higher preference for more intensive foods such as ripe cheese and fish in participants who came to the tasting compared to those that filled out the questionnaire without attending the tasting. Furthermore, we would like to mention that four of the tasting participants reported that they did not eat meat at the time of the tasting, but this did not significantly reduce their liking of the sauces containing meat; therefore, they were not excluded from the analysis (Table 3).

### 3.2. Overall Liking

The Hybrid sauce and the Meat sauce received the highest likings (mean value 6.9 ± SD 1.8 and 6.5 ± 1.8, respectively). Compared to the two meat-containing sauces, the Mealworm sauce obtained significantly lower likings (mean value 5.8 ± 1.7, *p* = 0.001) (Figure 2).

An analysis of variance was performed (ANOVA) and a significant product effect for the overall liking was observed of the three tested sauces (*p* < 0.0001). In order to conduct an analysis of the differences between the products, Tukey’s test (HSD) with a 95% confidence interval was performed. It was found that there was a significant difference in terms of liking between the Hybrid sauce and the Mealworm sauce (*p* < 0.0001) and the Meat sauce and the Mealworm sauce (*p* = 0.000), but not between the Meat sauce and the Hybrid Sauce (*p* = 0.170). 

### 3.3. CATA Analysis

For the analysis of the CATA data, the CATA results of the three products and those of the ideal product were included.

A penalty analysis was conducted with the liking data and the results of the CATA Analysis. From this, the must-have attributes as well as the must-not-have attributes could be identified. The must-have attributes for the ideal product had been identified for “reddish”, “juicy”, ”tasty”, “meaty”, “appropriate salt content”, “appropriate seasoning”, “tomato like- flavor”, “vegetable flavor”, and “appropriate intensive flavor”. In contrast, attributes which should not be present in the ideal product are “brownish” and “mushy”.

From the evaluation of the CATA data for the organoleptic characteristics, the highest positive effects on the liking score mean was seen for the attributes *schmackhaft* (tasty) and *passend intensives Aroma* (appropriate intense flavor/aroma). On the other hand, the attributes with the highest negative impact on the liking score mean were *bräunlich* (brownish) und *matschig* (mushy) (Figure 3).

Principal component analysis (PCA) (Figure 4) was generated using the significant CATA terms only (based on Cochran’s Q-Test, alpha 0.05). The first two dimensions explain more than 95% of the variance. It revealed that the ideal product was mainly described as having an “appropriate intense flavor” and further by the attribute “reddish”. The Meat sauce was mainly characterized by the attribute “meaty”. Moreover, the Mealworm sauce was particularly characterized by the attribute “brownish”; moreover, but not mainly, it was described as having a mushy consistency. Despite that, a clear description of the hybrid sauce was not made by the consumers. Taking a closer look at the data of the contingency table, we can see that characteristics such as “tasty”, “meaty”, and even “juicy” were common choices by the study participants to describe the Hybrid sauce; however, these attributes were frequently mentioned in terms of the other sauces (Meat sauce and ideal sauce).

The PCA points out that there is a clear distinction between the description of the Mealworm sauce and both the ideal and Meat sauce.

## 4. Discussion

The main goal of this research work was to find out how mealworm content influences the liking of developed pasta sauces. Product development with insects is a challenge [22,23,24] and requests strategies for decreasing visibility. To the best of our knowledge, there has been no results published on the development of a “meat like” pasta sauce with the partial and full replacement of mealworms. Other research has revealed that insects, like mealworms themselves, are associated with a low acceptance, although it has already been shown that with the increasing invisibility of the insects, e.g., as small pieces or powder, the acceptance of the animals themselves is increasing [21,22] and/or the fear of trying the insects is decreasing [33]. Furthermore, mealworm is considered the most common model for insects in Europe. The insect is easy to harvest and highest in terms of sustainability [22].

In our study, we used a shredded version of mealworms and revealed that the hybrid version (meat and mealworms) of the sauce was liked equally when compared to the commonly known sauce containing only meat (*p* > 0.05). The sauce, which only contained mealworms, was significantly less liked compared to both other sauces (*p* < 0.05). These results are comparable to a study where different burger patties were tested for acceptance [22]. We think that these are positive results with regard to the known lower acceptance of foods containing insects in Western countries [22]. That the hybrid and the meat sauce are equally liked is an important first step. The low visibility of the mealworms might have contributed to the acceptance or the liking, as well as the combination with a familiar product (similar to known products), as this has already been shown to simplify the hedonic response to unfamiliar foods [27,34].

Nevertheless, the overall mean values on a 9-point hedonic scale were below seven (e.g., 6.9, 6.7, and 5.8), so none of the sauces had very high ratings, although they did all have ratings above neutral. Also, the present absolute liking scores are similar to the abovementioned study [22]. We assume that several reasons may account for these results. The pasta sauces were tested in an unfamiliar test environment in a sensory booth that might have affected overall acceptance [35,36]. Furthermore, mushrooms were used in all three formulations, which might have further affected the acceptance of the meat sauce. However, mushrooms helped to provide a juicier consistency of the samples containing mealworms. This is also seen in the PCA (and when looking at the underlying contingency table), as the attribute juicy cannot be clearly assigned to a product. Moreover, a meat sauce is a well-known product, and one can assume very individual expectations as many have their own recipe. It was revealed in a statistic report about food consumption in Austria in 2022 that the so-called “Bolognese” meat sauce (with minced meat and tomatoes) was the most liked sauce (from a list of ten different ones) for 46% of the Austrian population. And for 57.6% of the study population, it was also their favorite pasta sauce [37]. To obtain insights here, the participants were asked to provide a response to the attributes that define their ideal sauces. Important attributes which resulted in a high liking or must-have attributes were “reddish”, “juicy”, “tasty”, “fleshy”, “appropriate intense flavor”, “appropriate intense seasoning”, “appropriate intense salt content”, “tomato-like flavor”, and “vegetable flavor”. On the contrary, the attributes “brownish” and “mushy” are considered undesirable, which, as shown in the PCA, are assigned to the “mealworm sauce”. These insights are very important for the further development of the sauces and innovations and provide insights into the preferences of Austrian consumers [38]. 

Overall, we can state that the product development of meat substitutes is still a challenge in terms of the imitation of meat flavor and textural attributes. Furthermore, a low sensory attractiveness regarding the product is an obstacle for its acceptance [34]. 

Although the study was planned very thoroughly, a few limitations should be stated here. Almost half of the participants who filled out the prior questionnaire did not attend the tasting, which might have influenced the acceptance of the mealworm sauces as these persons were described as more neophobic and possessed higher disgust sensitivity (Table 3). The refusal to participate in the tasting might also be a reason for why the mealworm sauce was still accepted (mean 5.8).

Furthermore, the mere exposure effect (contact to an unfamiliar product several times) might increase acceptance because of the familiarity of the product. Such was demonstrated by an educational sensory program published by Park and Cho (2015). The application of taste education brought about improvements, such as a significant decrease (*p* < 0.01) in the mean score of food neophobia, and the willingness to try new foods increased significantly (*p* < 0.001) [39]. Thus, it was demonstrated that applying educational programs for taste education improves food neophobia and willingness to try new foods. This is important to consider for future studies and for the market implementation of foods containing insects.

## 5. Conclusions

In this study, we tested products with different meat and mealworm concentrations. We saw that a partial substitution of meat by mealworms did not decrease liking for the sauce, but the complete substitution of meat with mealworms did. To increase consumer acceptance, further adaptation of the products in terms of (negative and positive) characteristics is proposed. One approach is to pay even more attention towards visual and consistency attributes. This further development could subsequently increase willingness to try insect products. Furthermore, scientific findings on the sensory aspects of insect products, leading to increased acceptance of insects as a whole and in processed form, should form the basis for further consumer-specific product developments.

## Figures and Tables

**Figure 1 foods-12-03202-f001:**
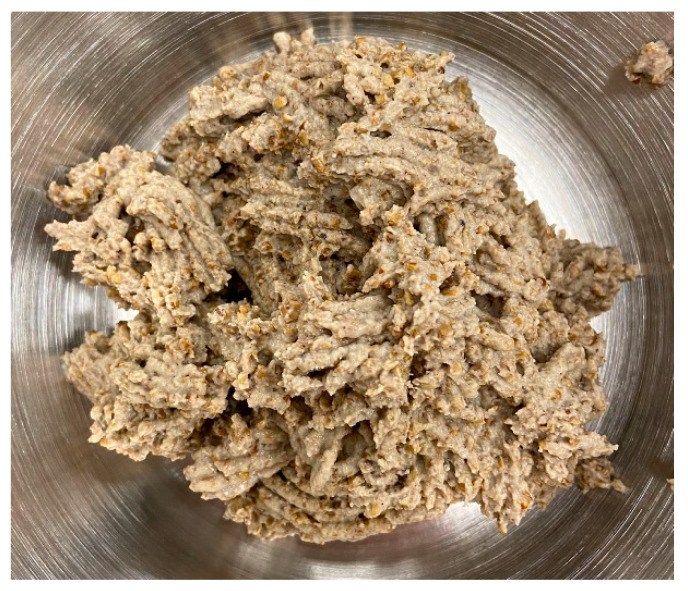
Minced mealworm after being processed through a meat grinder.

**Figure 2 foods-12-03202-f002:**
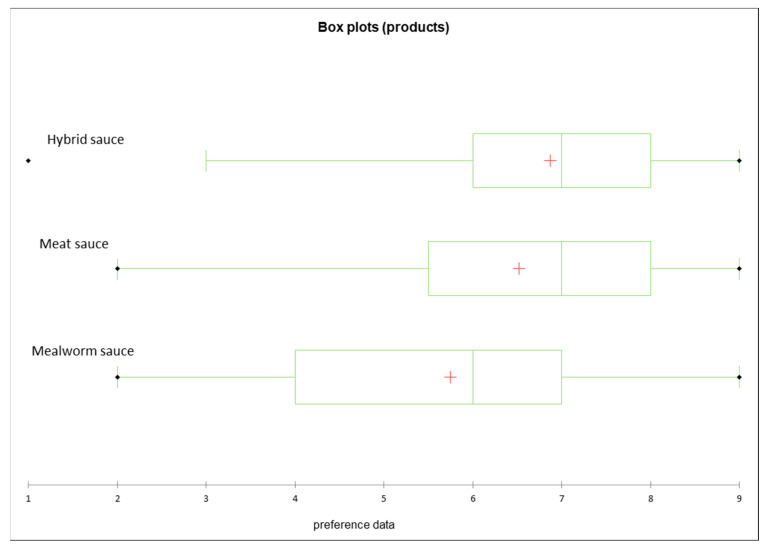
Box plots of the liking scores by product before centering. The red cross in the boxes marks the mean liking scores: Hybrid sauce 6.9, Meat sauce 6.5, Mealworm sauce 5.8. The *p*-value is considered significant at a value of <0.05.

**Figure 3 foods-12-03202-f003:**
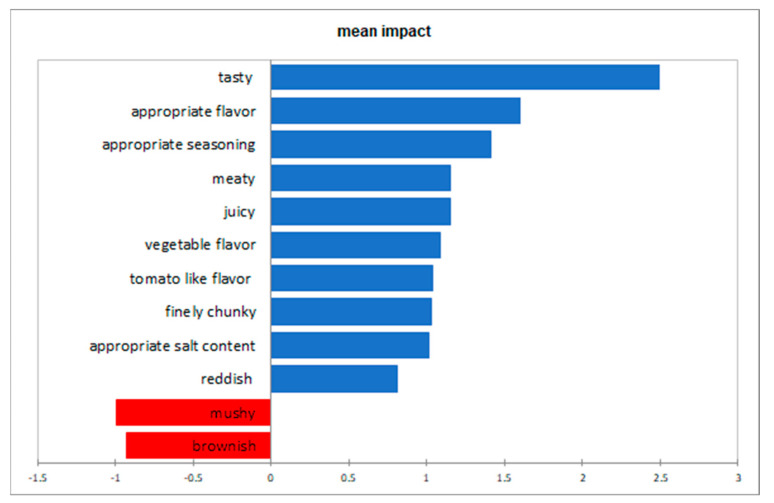
This figure shows the must-have attributes in blue (based on incongruence in which the attribute is missing in the real but not the ideal product), and the must-not-have attributes in red (based on the incongruence in which the attributes are missing the ideal but not the real product).

**Figure 4 foods-12-03202-f004:**
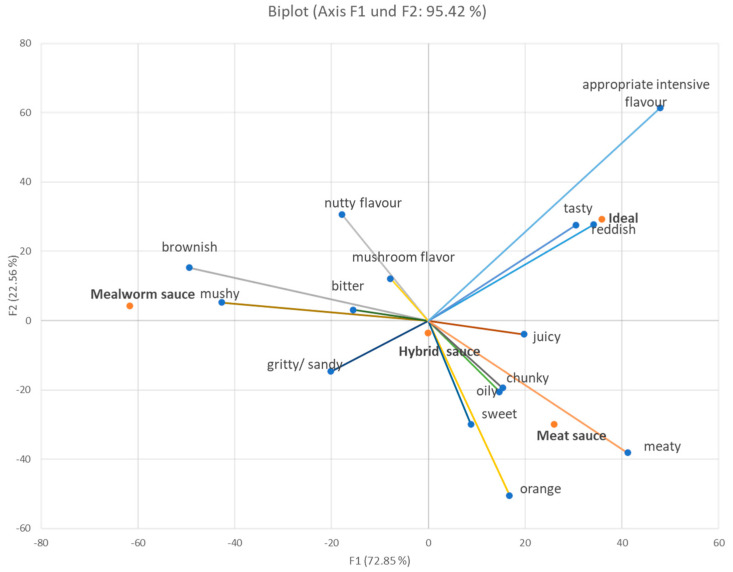
Sensory profile of the pasta sauces and the ideal product generated through principal component analysis (PCA) using the significant CATA terms only. The orange dots mark the active views (products), and the blue dots mark the active variables (attributes).

**Table 1 foods-12-03202-t001:** Protein content (in grams) in 100 g of the frozen worms, minced mealworms dough, and minced meat.

Protein Source	*Grams of Protein/100 g* *of the Worms or Meat*
*Frozen worms*	*18.94*
*Minced mealworms dough*	*13.74*
*Minced meat*	*18.00*

**Table 2 foods-12-03202-t002:** Total protein content in percentage of the three different sauces: Mealworm sauce, Meat sauce, and Hybrid sauce.

Pasta Sauces	*Protein Content from Minced Mealworm and/or Meat (%)*	*Protein Content from the Other Ingredients (%)*	Final Protein Content Product (%)
*Meat sauce—MS* *(meat only)*	*5.58*	*0.95*	*6.53*
*Hybrid sauce—HS* *(meat and mealworms)*	*4.92*	*0.95*	*5.87*
*Mealworm sauce—MWS* *(mealworms only)*	*4.26*	*0.95*	*5.21*

**Table 3 foods-12-03202-t003:** Participants characteristics.

Variable	All (*n* = 170)	Tasting Yes (*n* = 91)	Tasting No (*n* = 79)
Age	30 (23.0–43.3)	34 (26–46)	26 (23–37.8) *
Sex (male/female/other)	88/81	51/40	37/41/1
Food Neophobia	20 (16–25)	19 (15–23)	21 (17–28) *
Disgust overall_mean	2.9 (2.5–3.2)	2.9 (2.5–3.2)	3 (2.7–3.3) *
Disgust overall_sum	108 (95.7–120)	106 (91–117.5)	111 (100–123.8) *
Disgust oral	19 (16–21)	18 (15–20)	20 (17–23) *
Entomophagie score_mean	3.3 (3.1–3.7)	3.4 (3.2–3.7)	3.2 (2.9–3.6) *
Entomophagie score_sum	30 (28–33)	31 (29–33)	29 (26.3–32) *
Food pref for intensive foods	3.6 (3.3–4.1)	3.8 (3.4–4.2)	3.6 (2.9–4.1) *
Sports, min 8 h per week (yes/no)	33/137	17/74	16/63
Visit fitness studio regularly (yes/no)	44/126	22/69	22/57
Have you ever tried insects (yes/no)	101/69	62/29	39/40
Eating meat (yes/no)	153/17	87/4	66/13

Data shown as median (interquartile range), * *p*-value < 0.05 with Mann–Whitney U-Test between tasting yes and no.

## Data Availability

The data presented in this study are available on request from the corresponding author.

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
