# Peer review of "Liking and Description of Pasta Sauces with Varying Mealworm Content"

_foods, 2023, doi:10.3390/foods12173202_

Round 1

Reviewer 1 Report

Comments and Suggestions for Authors

The work plants the replacement of meat with edible insects in pasta sauce.

The article is well organised and interesting, however it can be improved according to the following opportunities for improvement:

- l54-56 the tone of the paragraph needs to be revised, as it reads somewhat exaggerated.

In section 2.1 it would be good to include what were the criteria for filtering the participants.

It is necessary to justify what is the contribution of figure 1.

in l180 the mixture of pork and beef is mentioned, but the proportions are not specified, the details of the formulations could be included in a table.

Add a heading to the column of table 1, it could be "Protein source".

What is the relevance of table 3, it would suffice to say that the concepts were translated from German into English.

In the CATA analysis section there is a comment in the manuscript, apparently from a previous revision, revise and delete.

It is not possible to see the letters in figure 4a correctly, please check the size.

The conclusion could be improved, with emphasis on the generation of knowledge in the sensory domain related to insect consumption.

Author Response

Dear Reviewer,

Thank you for reading our manuscript very carefully and for the helpful suggestions for improvement. You will find our response to your comments in the attached document.

Best regards,

Marlies Wallner

Reviewer 2 Report

Comments and Suggestions for Authors

Please see the comments listed below:

Introduction

1.       Lines 52-53: Old data are presented from a 2015 publication while more recent data can be found at https://www.fao.org/faostat/en/#data/FBS

2.       Lines 55-56: The reduction in meat consumption is recommended in the regions where the consumption is higher than required. Nevertheless, in countries that the consumption level of proteins from animal sources is much lower  (e.g., 1/8 or 1/5 of that of Western Europe), they do not need to reduce their meat consumption. Thus, the statement needs to be limited to those who consume more than their daily protein requirements.

3.       Line 62: The authors have discussed some of the advantageous of using insects but have not referred to potential health and safety concerns (e.g., the presence of heavy metals, their chitin content or its impact on the bioavailability of some nutrients) related to their consumption.

4.       Lines 71-72: Commas were used as decimal points.

Materials and methods:

1.       Lines 151-153: A run-on sentence

2.       Line 212: ‘noun’ or ‘known’?

3.       Lines 233-235: A run-on sentence

Results

1.       Table 4: The difference in the responses of ‘yes’ and ‘no’ groups for the ‘sports, min 8h per week’ question is considerable. Were they significantly different?

2.       Line 312: A new sentence starts from ‘Compared’.

3.       Table 5 is repetitive and can be removed by incorporating the information in the text.

4.       Section 3.2.1: Penalty analysis was not introduced in the data analysis section as one of the tests applied in the study. In addition, up to this section, the three treatment levels (sauces) were compared but suddenly, with no justification, the focus shifted to the whole data.

5.       The data from Table 6 can be incorporated into the text and Figure 4.

6.       In Figure 4, it is not clear why the results are presented in two separate sections (‘a’ and ‘b’).

7.       Line 370: ‘due to’ or ‘based on’?

8.       Line 371: Please clarify how the ‘ideal product’ data were incorporated into the PCA results.

9.       PCA results have not been discussed properly, and even the contributions of the two dimensions were not considered in presenting the findings of the test.

10.   Lines 375-376: A run-on sentence

Discussion:

The statements are repetitive with no focus on the actual findings of the study.

The items listed below are presented as examples, and unfortunately, the discussion section is difficult to follow in general and requires major revisions.

1.       Line 394: ‘with partly and fully replacement of’ or ‘the partial and full replacement of’?

2.       Lines 395-398: A run-on sentence that is not inline with the findings of the study. Disguising the mealworms in the current study did not result in increased acceptance or liking of the product.

3.       Lines 398-400: A run-on sentence

4.       Line 403: ‘equally’ instead of ‘equal’

5.       Line 404: what does ‘instead’ mean in this sentence?

6.       Lines 404-405: 'These results are comparable to a' instead of ‘These results a comparable to a study,’

7.       Lines 415-416: repetitive statements

8.       Line 420: ‘for a juicy consistency of’?

9.       Line 422: ‘very individual expectations’?

10.   Line 423: What does ‘some statistics reports’ mean? The sources need to be cited clearly in a scientific article.

11.   Line 436: extra apostrophe of its’

12.   Line 438: ‘filled out’ instead of ‘filled in’

13.   Line 448: A sentence started with a verb, without a subject.

Conclusion

The conclusion statements are not based on the findings of the results.

1.       Line 453: The timing of the inclusion of mealworm in pasta sauce was not tested in the current study thus referring to it by indicating that meat cannot be ‘immediately’ replaced by mealworm is not considered as an accurate statement.

2.       Higher acceptance of the hybrid sauce does not mean that “results suggest a stepwise substitution”. The experimental designs to test whether a stepwise substitution is preferred are different from what was done in this study. Thus, the conclusion statement is not based on the results.

Comments on the Quality of English Language

Please see the detailed comments above.

The discussion section requires extensive revisions.

Author Response

(The authors gave the same response as above.)

Round 2

Reviewer 1 Report

Comments and Suggestions for Authors

I have reviewed the manuscript and the authors have responded satisfactorily to the comments made by this reviewer. It is recommended that the paper be accepted in its present form.

Reviewer 2 Report

Comments and Suggestions for Authors

I enjoyed reading your revised article. Thank you.

Comments on the Quality of English Language

No further comments